# Pattern of Regulatory T Cells, Resident Memory T Cells, and Exhausted T Cells in Human Pericardial Fluid Samples of Cardiovascular Patients

**DOI:** 10.3390/ijms26209852

**Published:** 2025-10-10

**Authors:** Barbara Érsek, Júlia Opra, Nóra Fekete, Mandula Ifju, Viktor Molnár, Edina Bugyik, Éva Pállinger, Andrea Székely, Tamás Radovits, Béla Merkely, Edit I. Buzás

**Affiliations:** 1Institute of Genetics Cell- and Immunobiology, Semmelweis University, 1085 Budapest, Hungary; osku.opra.julia@semmelweis.hu (J.O.); fekete.nora@semmelweis.hu (N.F.); bugyik.edina@semmelweis.hu (E.B.); pallinger.eva@semmelweis.hu (É.P.); buzas.edit@semmelweis.hu (E.I.B.); 2Doctoral School, Semmelweis University, 1085 Budapest, Hungary; ifju.mandula@semmelweis.hu; 3Institute of Genomic Medicine and Rare Disorders, Semmelweis University, 1083 Budapest, Hungary; molnar.viktor@semmelweis.hu; 4Department of Anesthesiology and Intensive Therapy, Semmelweis University, 1082 Budapest, Hungary; szekely.andrea1@semmelweis.hu; 5Department of Cardiology Heart and Vascular Center, Semmelweis University, 1085 Budapest, Hungary; radovits.tamas@semmelweis.hu (T.R.); rektor@semmelweis.hu (B.M.); 6HCEMM-SU Extracellular Vesicle Research Group, 1089 Budapest, Hungary; 7HUN-REN SU Translational Extracellular Vesicle Research Group, 1089 Budapest, Hungary

**Keywords:** immune, heart, transplantation, T cells, pericardial fluid

## Abstract

This study investigates T cell subsets in pericardial fluid samples obtained from heart transplantation donors, heart transplantation recipients, and coronary artery bypass graft patients. Using flow cytometry, we characterized regulatory T cells (Tregs), tissue-resident memory T cells (Trm), and exhausted T cells based on specific markers. Our results showed significant alterations in the CD4^+^ and CD8^+^ T cell subsets, migration (CXCR3, CCR5), and exhaustion markers (PD-1, TIM3) across the groups. Notably, Tregs and Trm cells were enriched in recipients, while markers of T cell exhaustion showed a complex regulation. These findings provide novel insights into the local immune regulation in cardiac disease and transplantation.

## 1. Introduction

T cells are vital to the immunological response of the heart and have a role in both pathogenic and protective responses. It is known that the heart is an active site of immune cell infiltration during infection, injury, and chronic diseases (such as heart failure, myocarditis, and ischemia) [1,2,3,4]. The focused distribution of T cells in the heart (cardiotropism) is a significant feature of the above conditions.

Chemokine receptors, which mediate the preferential migration of T cells to inflammatory or damaged heart tissue, are the primary regulators of T cell cardiotropism. Activated T cell-expressed CCR5 interacts with chemokines like CCL5 (RANTES), important for T cell recruitment and induced in the heart during inflammation [5]. CXCL9, CXCL10, and CXCL11 are ligands of CXCR3, another receptor that is strongly expressed on Th1 cells and cytotoxic T cells [6]. These chemokines, which direct T lymphocytes to the heart, are frequently produced in response to infection or cardiac damage [7,8]. Furthermore, the extravasation of T lymphocytes from the circulation into the heart tissue is facilitated by adhesion molecules such as ICAM-1, VCAM-1, and selectins expressed on the vascular endothelium [9,10]. Integrins, including LFA-1 and VLA-4, further support T cell binding and migration across endothelial barriers [11,12].

Dilated cardiomyopathy is a progressive disease defined by ventricular dilatation and impaired systolic function [13,14]. The potential role of T cells as contributors to disease pathophysiology has emerged from their participation in a series of inflammatory events that could lead to disease progression. Although dilated cardiomyopathy may result from genetic [15,16] and infectious causes, as well as from the effect of toxins [17], immune-mediated mechanisms are emerging as important factors in disease development. The participation of T cells, especially autoreactive T cells, has been implicated in the chronic inflammation that enhances myocardial damage in DCM [18,19].

T cells in DCM may target cardiac autoantigens, leading to sustained inflammation and myocardial remodeling [18,20]. Autoantibodies to proteins such as cardiac myosin [15,21] or titin, which are associated with T cell responses, may contribute to the immune-mediated injury observed in some forms of DCM. This suggests that T cell cardiotropism in DCM may be driven by molecular patterns similar to those observed in autoimmune myocarditis.

Chronic ischemic heart disease, the main indication for coronary artery bypass graft (CABG), is characterized by a prolonged imbalance between myocardial oxygen supply and demand, leading to tissue damage, chronic inflammation, and immune activation [22]. Chronic ischemia creates a pro-inflammatory environment that promotes the recruitment and activation of T cell subsets. CD4^+^ and CD8^+^ T cells have been shown to contribute to myocardial injury and vascular remodeling through cytokine release and direct cytotoxic effects [23,24,25].

Beyond providing friction-bearing and mechanical support, pericardial fluid serves as a reservoir for immune cells and other bioactive substances that can be dysregulated in various disease conditions [26,27,28]. The moderate fluid turnover rate makes the pericardial fluid a potentially informative biofluid to assess heart disorders. Although the location of the pericardial fluid does not allow for easy diagnostic sampling, heart transplantation provides a unique opportunity, allowing investigators to obtain this uncommon biological sample at the same time from both donors (who do not have heart disease) and recipients (who have heart failure of various causes).

In the heart, regulatory T cells (Tregs) are involved in controlling inflammation and promoting tissue repair [29,30]. Studies have shown that Tregs suppress excessive immune responses that can otherwise lead to tissue damage and fibrosis [31]. CCR4 on Tregs binds to chemokines such as CCL17 and CCL22 produced by inflamed or damaged tissues [32]. It enhances the ability of Tregs to migrate towards cardiac tissue undergoing stress or injury, e.g., during myocardial infarction or myocarditis. On the other hand, a recent study showed that CCL17 enhanced cardiac injury because of the suppression of Treg migration [33]. CCR4-expressing Tregs exhibit enhanced suppressive functions compared to other subsets [34,35]. CD45RA on Tregs functions as a marker of naive Tregs with a stable, long-lived phenotype. These cells have lower immediate suppressive capabilities compared to effector Tregs, but play a crucial role in maintaining immune tolerance and can differentiate into more suppressive effector Tregs when needed [36]. CD45RA^+^ Tregs provide a reservoir of regulatory cells that can respond to new antigens and maintain peripheral tolerance over time [37].

Tissue-resident memory T cells (Trms) reside permanently in non-lymphoid organs, without ever moving back into the bloodstream. In contrast to circulating T cells, they deliver quick immune responses when they come into contact with pathogens or stress signals in the tissue microenvironment [38]. CD103 promotes tissue retention by interacting with epithelial cadherin, a molecule commonly used to identify CD8^+^ Trms [39]. The presence of Trms was observed in the epicardial adipose tissue in patients with atrial fibrillation [40], and their contribution to injury-induced myocarditis was also discussed in mice. Moreover, it was found that compared to the peripheral blood, the CD69^+^ Trm subpopulation was enriched in the pericardial fluid of patients with different pathologies [41]. CD49a helps Trms adhere to the extracellular matrix, promoting their retention in tissue sites. CD49a^+^ CD8^+^ Trm cells have been shown to possess heightened cytotoxic capabilities compared to their CD49a- counterparts [42]. They are involved in balancing immune protection and tissue inflammation. While they are potent responders to infections, they maintain a controlled immune presence that avoids excessive tissue damage during inflammation [43,44].

Chronic infections and cancer are commonly linked to exhausted T cells (Tex), which lead to a dysfunctional state marked by decreased cytokine output and proliferative capacity driven by persistent antigen stimulation [45,46]. Their compromised function results from the high expression of inhibitory receptors such as PD-1, LAG-3, and TIM-3. The exhaustion can be harmful by reducing the effectiveness of virus removal or tumor suppression, but it can also be protective by limiting tissue damage caused by overly aggressive immune responses [47].

The aim of this study was to characterize the distribution and functional phenotypes of Tregs, Trms, and Tex in the pericardial fluid of heart transplantation donors, recipients, and CABG patients. Specifically, we assessed chemokine receptor-driven inflammatory recruitment (CCR5, CXCR3), regulatory and differentiation states of Tregs (CCR4, CD45RA), homing and retention features of Trm cells (CD49a), and the balance between transient and terminal T cell exhaustion (PD-1, TIM-3). By systematically comparing these immune subsets across the three patient groups, our goal was to identify disease- and transplantation-specific immune patterns that may underlie cardiac inflammation, immune regulation, and clinical outcome variability.

## 2. Results

### 2.1. Enrichment of Inflammatory-Homing T Cells in Transplantation Recipients

In the context of T cell cardiotropism, where CCR5 and CXCR3 [5,7,8] are among the key receptors guiding migration into the inflammatory environment of the heart, the expression of these receptors was analyzed in pericardial fluid samples from both CD4^+^ and CD8^+^ T cells. The percentages of chemokine receptor-positive cells were compared across patient groups.

Within both the CD4^+^ and CD8^+^ T cell subsets, distinct patterns were observed between patients who had undergone cardiac transplantation and those with CABG.

A reduced frequency of CD4^+^ T cells was detected in the CABG patient group compared to all other groups (Figure 1A). However, the proportion of CD4^+^ T cells with an inflammatory-homing phenotype did not differ significantly between the CABG and donor groups, suggesting no selective enrichment in CABG hearts (Figure 1B). In contrast, while in the transplantation recipient group the total CD4^+^ T cell frequency remained unchanged compared to the donor group (Figure 1A), a significant increase in CD4^+^ T cells expressing inflammatory trafficking receptors was observed (Figure 1B). The selective enrichment of CCR5^+^ CXCR3^+^ CD4^+^ T cells in transplant recipients suggests that these represent inflammatory effector phenotypes. Similar CCR5^+^ CXCR3^+^ T cells have been previously described as dominant populations in transplant-associated inflammation [48]. This further supports the interpretation that the pericardial immune milieu in recipients is characterized by an inflammatory T cell bias, distinct from the global CD4^+^ loss observed in CABG patients. No similar pattern was identified within the CD8^+^ T cell subset.

### 2.2. Increased Infiltration but Decreased Treg Function in Transplant Recipients

Tregs are involved in the regulation of inflammation in the heart; thus, Treg distribution and characteristics were analyzed within the study groups.

While no significant difference was observed in the overall frequency of Tregs, a higher number of CCR4^+^ Tregs was detected in the transplantation recipient group compared to donors (Figure 2A). Given that CCR4^+^ Tregs are more suppressive compared to other Treg subsets, their presence is highly relevant in limiting injury during cardiac stress and the course of chronic heart disease through the regulation of inflammation. This pattern seems to be specific to transplantation recipients, since no such difference was observed in the CABG group (Figure 2A).

Interestingly, the subset of CD45RA-expressing Tregs was significantly reduced in transplantation recipients compared to CABG patients (Figure 2B). Upon antigen recognition, Tregs lose CD45RA expression and differentiate into effector Tregs with increased suppressive capacity [36].

Thus, the low representation of CD45RA^+^ Tregs and the elevated proportion of the highly suppressive CCR4^+^ Tregs in the recipient group suggest increased Treg activation and highlight a functional difference between the two disease groups (Figure 2A,B). The combination of fewer CD45RA^+^ (naive) Tregs with more CCR4^+^ (effector-skewed) Tregs in recipients supports a state of enhanced regulatory activation but a contracted naive reservoir, a configuration that may provide short-term control of inflammation yet limit long-term regulatory adaptability.

### 2.3. Tissue Resident Memory T Cells (Trms) Are Enriched in Patients with DCM, but Their Homing and Retention May Be Impaired in the Pericardial Fluid

There is very limited data regarding resident immune cells in pericardial tissue, despite their potentially important role in regulating tissue homeostasis. For this reason, the presence of Trm cells was studied.

The proportion of Trm cells was significantly higher in transplantation recipients compared to CABG patients (Figure 3A). Further analysis of functional subsets revealed a marked reduction in CCR5^+^ CXCR3^+^ Trm cells in both CABG patients and recipients compared to donors (Figure 3B), indicating a potential impairment in the homing/immigration capacity of Trm cells in pathological conditions (CABG and heart failure). CD49a helps in the retention of Trm cells, and CD49a Trms have a greater cytotoxic capacity. The CD49a^+^ Trm subset, associated with tissue retention and survival, was significantly reduced in CABG and recipient groups relative to donors (Figure 3C).

Thus, while transplantation recipients displayed higher total Trm frequencies, the relative reduction in CCR5^+^ CXCR3^+^ and CD49a^+^ Trm suggests that these resident pools may be numerically expanded but functionally impaired in their ability to home and remain in cardiac tissue, potentially limiting protective tissue surveillance.

### 2.4. Reduced Number of Transiently Exhausted but Increased Number of Terminally Exhausted T Cells in the Cardiac Patients

In chronic inflammatory conditions such as myocarditis, autoimmune diseases, or other diseases that affect the cardiac tissues, the repeated/sustained exposure to antigens may result in T cell exhaustion in cardiac tissues.

The examination of Tex cells showed no difference in the CD4^+^ or CD8^+^ and PD-1^+^ subpopulations. However, a marked decrease was observed in the CCR5^+^ CXCR3^+^ PD-1^+^ cell number in both CD4^+^ and CD8^+^ fractions in the recipient patient group (Figure 4A,B).

To further analyze T cell exhaustion, TIM3 expression was studied. Interestingly, we found that in the case of the CABG patients, the CD4^+^ CCR5^+^ CXCR3^+^ PD-1^+^ T cells expressed more TIM3 molecules on their surfaces (Figure 4E). Thus, although the percentage of TIM3^+^ T cells alone did not differ between the subject groups (Figure 4C,D), the expression level on each cell was higher with PD-1^+^. Taken together, these findings indicate that CABG patients accumulate more terminally exhausted (PD-1^+^TIM-3^+^) T cells, consistent with chronic ischemic antigenic drive, whereas transplant recipients show fewer exhausted subsets, likely reflecting both immunosuppression and continuous T-cell turnover.

### 2.5. Clinical Variables

Among transplant recipients, no significant association was found between T-cell levels and the complications, clinical scores, echocardiographic parameters, hemodynamic measurements, or laboratory values detailed in Table 1 (Appendix A).

In the CABG patient group, no significant associations were observed between T-cell levels and baseline comorbidities or preoperative risk scores. A positive correlation was found between creatine kinase and Trm CD8^+^/CCR5^+^ CXCR3^+^ levels (r = 0.976; *p* < 0.001; q = 0.016). Unfortunately, CK-MB data were not available for these patients. Moreover, a negative correlation was observed between TAPSE and Treg CD4^+^FOXP3^+^CCR5^+^Q4:CCR4-, CXCR3- levels (r = −0.991; *p* < 0.001; q = 0.016), suggesting that impaired right ventricular systolic function may be associated with increased numbers of these Treg cells. However, the interpretation of these results is limited by the fact that all CABG patients had TAPSE values within the normal range (median 22.00 mm, IQR: 20.00–24.00 mm). Further studies that include a larger cohort of patients with both normal and reduced right ventricular systolic function are required to clarify this relationship.

## 3. Discussion

This paper comprehensively examines T-cell phenotypes in the pericardial fluid of heart transplantation donors, recipients, and patients undergoing coronary artery bypass grafting.

Pericardial fluid lymphocytes may originate from the myocardium and enter the fluid via drainage pathways, while others may migrate toward the heart under inflammatory chemokine gradients. Animal studies have shown that Gata6^+^ pericardial macrophages can relocate into injured myocardium [4], supporting the concept of dynamic immune exchange between the pericardial space and cardiac tissue.

The lower frequency of CD4^+^ T cells in CABG patients relative to donors and recipients suggests a general dysregulation of T-cell responses in ischemic heart disease, indicating disturbed immunological homeostasis. However, the percentage of CXCR3^+^CCR5^+^CD4^+^ T cells in recipients was significantly higher than in the other groups. Given that Th1 cell trafficking and retention are indicated by CXCR3 and CCR5 [49], our finding points to an increased immigration and a Th1-shifted immune response in the failing heart. The typically higher amounts of chemokines CXCL9, CXCL10, and CCL5 that attract CXCR3^+^CCR5^+^ cells (which are detected in failing hearts [50] may support this theory. In line with our observations, Iskandar et al. demonstrated increased infiltration of inflammatory cells and elevated levels of inflammatory cytokines in the pericardial fluid of patients with chronic heart failure. This reflects T cell activation and highlights the contribution of systemic inflammation to disease progression [51].

Taken together, these findings suggest that ischemic heart disease drives a broad depletion of CD4^+^ cells, weakening the adaptive immune repertoire, while end-stage heart failure is associated with preserved CD4^+^ numbers but selective enrichment of CCR5^+^CXCR3^+^ inflammatory subsets. This divergence points to different immune pressures: ischemia leading to attrition and exhaustion, while chronic heart failure promotes persistence of inflammatory phenotypes within the pericardial space.

The significantly higher number of the more suppressive CCR4^+^ Tregs in recipients compared to donors and CABG patients may represent a compensation mechanism for excessive inflammation and tissue damage. This is further supported by the observation that recipients had fewer CD45RA^+^ Tregs, consistent with a shift from naïve to activated or memory-like regulatory T cells. On the other hand, the reduced CCR4^+^ Treg levels in CABG patients compared to recipients may point to a distinct immunological dynamic that is most likely characterized by less localized immunity control. Alexander et al. reported decreased circulating regulatory T-cell pools in human hypertension, suggesting that in certain cardiovascular conditions, the blood compartment is relatively depleted, either because of impaired survival or redistribution to inflamed tissues [52]. Taken together, pericardial enrichment and the hypertension-related circulating deficit suggest that cardiovascular inflammation shapes the distribution of T-cell subsets between blood and cardiac tissues, depending on disease context and chemokine signals.

This redistribution underscores that pericardial fluid is not merely a passive ultrafiltrate but reflects active local immune remodeling. The accumulation of effector Tregs in failing hearts may provide short-term restraint of inflammation, but the loss of naive precursors could compromise long-term regulatory stability. Conversely, the relative preservation of naive Tregs in ischemic patients suggests a more balanced regulatory repertoire despite the chronic inflammatory milieu.

Compared to CABG patients, transplantation recipients had considerably higher levels of Trms. This finding is consistent with the observations regarding Tregs and further emphasizes the importance of such cell types in contributing to homeostatic processes in the failing heart. The persistent presence of these cells in the end-stage heart is probably caused by tissue remodeling, chronic inflammation, and hypoxia. The lower frequency of CXCR3^+^CCR5^+^ Trm cells in the transplantation recipients and CABG patients relative to donors, however, suggests potential defects in chemokine-mediated Trm localization. This decrease in CABG patients most likely reflects decreased T-cell trafficking, whereas in cardiac transplant recipients, it may be the consequence of long-term immunological dysregulation. Beyond numerical changes, cardiac-adjacent Trm can exert functional effects on the myocardium: human epicardial adipose tissue harbors Trm that modulate atrial cardiomyocyte electrophysiology by altering Ca^2+^ flux and activating inflammatory/apoptotic pathways, particularly at the inflamed epicardial adipose tissue border in AF [40]. In parallel, pericardial/cardiac Trm expand after injury and can drive myocarditis in experimental models, with human pericardial macrophages providing IL-15 that supports Trm maintenance [41]. These data reinforce that Trm within pericardial or epicardial niches can influence myocardial inflammation and, context-dependently, electrophysiologic vulnerability. Our observation of increased but phenotypically altered Trm in end-stage hearts suggests that chronic damage promotes Trm persistence while impairing their retention and tissue-protective functions. This maladaptive reshaping may weaken local immune surveillance and allow progressive fibrosis or arrhythmogenic remodeling. In CABG patients, the reduced Trm pools together with terminally exhausted T cells may further compromise repair and adaptation.

PD-1^+^ T cells are in a transiently or moderately exhausted state and can often regain function if PD-1 signaling is blocked. However, PD-1^+^ TIM-3^+^ double-positive T cells are typically in a more profoundly exhausted state, where TIM-3 adds an extra layer of inhibition, leading to severe functional impairment [53]. The observed patterns of T cell infiltration and activation in the recipient group likely reflect active effector T cell trafficking and continuous fluctuations in the number of heart-directed T cells. Strikingly, there was a lower number of exhausted T cells. TIM-3 expression on the PD-1^+^ cells was considerably higher in CABG patients than in transplant donors within the CD4^+^CXCR3^+^CCR5^+^PD-1^+^ T-cell population. In the setting of ischemic heart disease, the increased TIM-3 expression in CABG patients may be indicative of terminal exhaustion. This dichotomy indicates that chronic ischemia drives T cells into terminal dysfunction, which may impair repair mechanisms and foster maladaptive remodeling, while end-stage failure is characterized by preserved effector activity but skewed toward inflammatory programs. These contrasting immune landscapes provide biological plausibility for the different clinical trajectories of ischemic vs. non-ischemic failing hearts.

Although our study provides the first detailed characterization of regulatory, resident, and exhausted T-cell subsets in human pericardial fluid across distinct clinical contexts, several aspects remain unresolved. The study design limits the ability to determine temporal dynamics, such as whether the observed inflammatory enrichment in end-stage hearts represents a driver of progression or a consequence of prolonged cardiac injury. Likewise, we cannot yet clarify to what extent pericardial immune cells actively interact with myocardial tissue versus reflecting systemic inflammation. Despite these uncertainties, the pericardial compartment offers a unique immunological space, directly adjacent to the diseased heart, which is rarely accessible in living patients. We also acknowledge some technical limitations. The sample size was relatively small, and long-term follow-up data were not available for the CABG patients. Furthermore, we did not obtain information on the cellular composition or the average cell count of the pericardial fluid; however, such data have been reported in other publications across the literature [54].

Future work should therefore move beyond phenotyping toward functional and integrative approaches to establish whether T-cell states correspond to histological inflammation, fibrosis, or remodeling. Clinically, prospective studies linking pericardial immune profiles with postoperative outcomes, ventricular recovery, or arrhythmia would clarify their prognostic value. Ultimately, dissecting pericardial immunity may identify cellular and molecular signatures that serve as biomarkers of cardiac damage and may reveal novel immunomodulatory targets in advanced heart disease.

## 4. Materials and Methods

### 4.1. Study Design, Setting, and Participants

The study design and sample processing steps are summarized in Figure 5. Well-characterised pseudonymised human pericardial fluid samples were obtained from 12 donors, 15 recipients, and 11 CABG patients. These samples were obtained from the Transplantation Biobank of the Heart and Vascular Centre at Semmelweis University, Budapest, Hungary. The analysis was carried out at the Institute of Genetics, Cell- and Immunobiology. The procedure of sample procurement was reviewed and approved by the institutional and national ethics committee (ethical permission numbers: ETT TUKEB 7891/2012/EKU [119/PI/12] and ETT TUKEB IV/10161-1/2020). Patient clinical data were collected from the Transplantation Biobank database. Our study was conducted in accordance with Eurotransplant standards for organ sharing and with the Hungarian National Blood Transfusion Service.

### 4.2. Variables and Definitions

Clinical data of CABG patients and HTx (heart transplantation) recipient variables were collected from electronic medical records available in our institutional database. The demographic data of subjects are summarized in Table 1. The median age of the donors was 39.50 (26.50–48.50) years, and 11 (91.7%) donors were male. The causes of death were intracerebral hemorrhage in 58.3% (*n* = 7), traumatic subdural hemorrhage in 8.3% (*n* = 1), diffuse brain injury in 25.0% (*n* = 3), and anoxia in 8.3% (*n* = 1) (Table 2). Primary graft dysfunction (PGD) was defined according to the consensus criteria established by the International Society of Heart and Lung Transplantation (ISHLT). The decision to initiate mechanical circulatory support (MCS) was made by a multidisciplinary team (including a cardiologist, cardiac surgeon, and cardiac anaesthesiologist) based on the international guidelines [55]. Acute rejection was defined as an event requiring escalation of immunosuppressive therapy, marked by an endomyocardial biopsy result of ≥2R based on the ISHLT grading system or a non-cellular rejection associated with hemodynamic instability. Post-transplant vasoplegia was diagnosed using binary criteria: a cardiac index ≥2.5 L/min/m^2^ in combination with a requirement for vasopressor support (noradrenaline ≥5 µg/min, adrenaline ≥4 µg/min, or vasopressin ≥1 unit/h) to maintain a mean arterial pressure of 65 mmHg for at least six continuous hours within the first 48 h postoperatively [56]. Because of the limited sample size, we were not able to assess differences in the case of rejection, reoperation, mortality, and postoperative MCS. Due to the limited sample size, we calculated the Index for Mortality Prediction After Cardiac Transplantation (IMPACT) [57] and the United Network for Organ Sharing (UNOS) scores for transplant recipients [58], while the European System for Cardiac Operative Risk Evaluation (EuroSCORE) [59] and Canadian Cardiovascular Society (CCS) scores [60] were used for patients underwent CABG. Available data on donors are summarized in Table 2.

### 4.3. Sample Collection and Preparation

ACD A (Acid Citrate Dextrose) vacutainer tubes (BD Vacutainer System, BD Biosciences, San Jose, CA, USA) were used for the collection of pericardial fluid samples after sternal splitting and opening of the pericardium. We collected 5 mL of pericardial fluid from each patient. Cells were collected via centrifugation (300× *g*, 10 min, 4 °C) and were frozen in a solution containing 900 µL fetal bovine serum and 100 µL DMSO. The samples were stored at −80 °C for 24 h and then transferred to liquid nitrogen until measurement.

### 4.4. Flow Cytometry

For flow cytometry surface staining, mononuclear cells from pericardial fluid were labeled with the following antibodies in various combinations: CCR5–APC, CXCR3–APC/Cy7, CCR4–PE/Cy7, FOXP3–PE, CD103–PE, CD8–PerCP/Cy5.5, CD69–FITC, CD25–Pacific Blue, PD-1–Pacific Blue (Sony Biotechnology Inc., San Jose, CA, USA), CD4–BUV395, CD49a–BUV395 (BD Biosciences), Tim-3–BV605, CD45RA–BV605, and fixable Live/Dead Scarlet (Thermo Fisher, Waltham, MA, USA).

Cell viability was assessed using fixable Live/Dead Scarlet (Thermo Fisher) according to the manufacturer’s instructions, and dead cells were excluded from any further analysis. The general gating strategy to exclude dead cells is shown in Figure 6. Subsets of T cells were defined by sequential gating from the lymphocyte population, and their levels are reported as relative frequencies within the appropriate parent populations. Accordingly, the data reflect proportional distributions instead of absolute cell counts.

For the detection of intracellular antigens, Foxp3/Transcription Factor Staining Buffer Set (Thermo Fisher) was used following the manufacturer’s instructions. Briefly, Fc blocking was performed using 10% heat-inactivated human serum (Sigma-Aldrich, St. Louis, MO, USA) for 10 min prior to immune staining. Cells were labelled for viability, washed, stained for surface antigens, fixed with the fixative agent of the kit, permeabilized with the Perm/Wash solution of the kit, washed, stained for intracellular antigens for 1 h at 4 °C, washed, and resuspended in MACS buffer (Miltenyi, San Jose, CA, USA) for flow cytometry measurement.

Raw data were analyzed with FlowJo v.10.10.0. Regulatory T cells (Treg) were defined by CD4^+^CD25^+^Foxp3^+^ expression, CD8^+^CD103^+^CD69^+^ cells as tissue-resident memory T cells (Trm), and both CD4^+^ and CD8^+^ PD-1 and/or TIM-3^+^ as exhausted T cells (Tex). Cells double positive for CCR5 and CXCR3 were considered as inflammatory-homing T cells compatible with the inflammatory environment of the heart.

### 4.5. Statistics

Data analysis was carried out with GraphPad v 9.11 (GraphPad Software, Inc., San Diego, CA, USA). Outliers were identified using the Outlier removal method. One-way analysis of variance (ANOVAs) was used, and Tukey’s multiple-comparison test was used as a post hoc test. Clinical data are reported as medians with interquartile ranges (IQR; 25th–75th percentiles). Group differences were evaluated using the non-parametric Mann–Whitney U test for continuous variables. Associations between clinical outcomes and levels of T cells were evaluated using Spearman’s correlation to assess differences in T-cell levels across risk score categories, and one-way analysis of variance on ranks was applied. Following the statistical tests for clinical variables, the Benjamini–Hochberg procedure was used to control the false discovery rate (FDR) at 5%, applied separately for the CABG and heart transplant recipient cohorts. In all analyses, *p* < 0.05 was considered statistically significant, and individual values and mean +/− SEM were visualized.

## 5. Conclusions

In summary, our study demonstrates that the immune landscape of the pericardial space differs markedly between end-stage heart failure, ischemic heart disease, and healthy controls. While ischemia was associated with global depletion and terminal exhaustion of T cells, failing hearts retained overall numbers, but were skewed toward inflammatory and resident phenotypes with altered regulatory balance. These divergent patterns underscore that pericardial immunity reflects disease-specific remodeling rather than a uniform response. Ultimately, dissecting pericardial immune alterations may not only advance our understanding of cardiac pathophysiology but might also provide novel biomarkers and therapeutic targets for advanced heart disease.

## Figures and Tables

**Figure 1 ijms-26-09852-f001:**
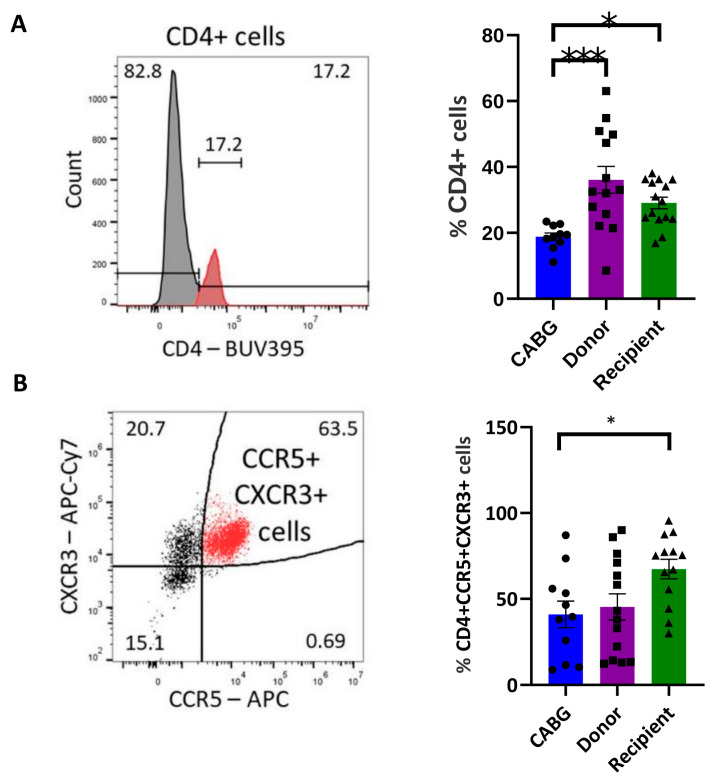
**Distribution of CD4^+^ T cells associated with inflammatory recruitment.** Pericardial fluid samples were stained and analyzed by flow cytometry and the percent of positive cells for CD4^+^ T cells and CXCR3^+^ CCR5^+^ inflammatory homing T cells was compared between the individual groups: (**A**) one-way ANOVA, *** *p* < 0.001; Tukey’s multiple-comparison test * *p* < 0.05; CABG *n* = 10, donor *n* = 14, recipient *n* = 15; (**B**) one-way ANOVA, * *p* < 0.05; Tukey’s multiple-comparison test * *p* < 0.05; CABG *n* = 11, donor *n* = 14, recipient *n* = 13. Representative flow cytometry images are shown.

**Figure 2 ijms-26-09852-f002:**
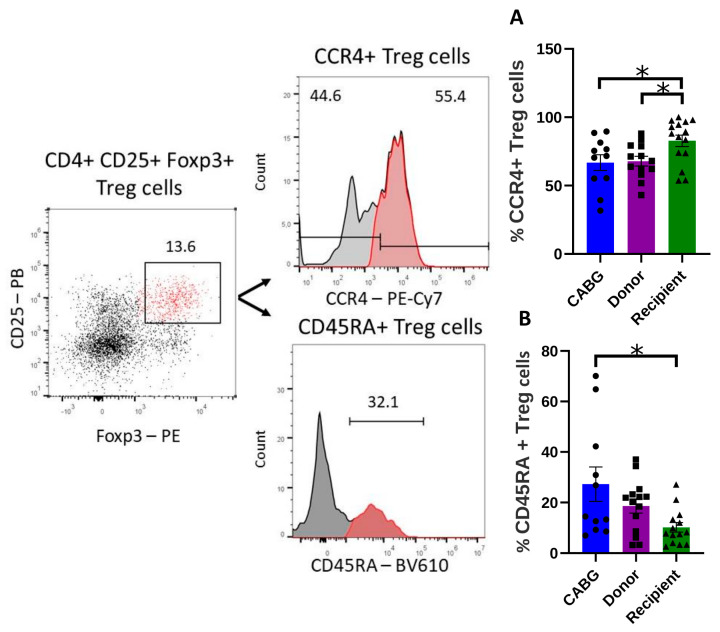
Regulatory T cell patterns in the disease groups: (**A**) CCR4^+^ inflammatory-homing phenotype regulatory T cell percentage was analyzed between the groups with flow cytometry (one-way ANOVA, * *p* < 0.05; Tukey’s multiple-comparison test * *p* < 0.05; CABG *n* = 11, donor *n* = 13, recipient *n* = 15); (**B**) the expression of CD45RA on Treg cells was compared between the groups (Tukey’s multiple-comparison test * *p* < 0.05; CABG *n* = 11, donor *n* = 14, recipient *n* = 14). Representative flow cytometry images are shown.

**Figure 3 ijms-26-09852-f003:**
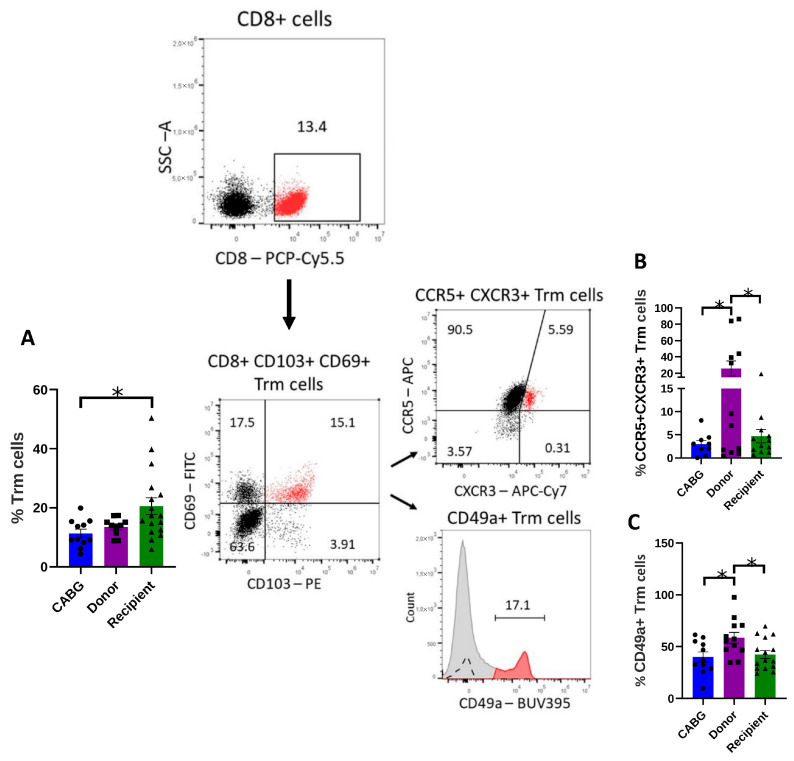
**Tissue resident memory T cell distribution:** (**A**) The expression of CD8^+^ CD103^+^ CD69^+^ antigens was measured with flow cytometry, and the percentages of the expressing cells are demonstrated (one-way ANOVA, * *p* < 0.05; Tukey’s multiple-comparison test * *p* < 0.05; CABG *n* = 11, donor *n* = 12, recipient *n* = 17); (**B**) CCR5^+^ CXCR3^+^ Trm cells relevant to cardiac inflammatory recruitment were evaluated (one-way ANOVA, * *p* < 0.05; Tukey’s multiple-comparison test * *p* < 0.05; CABG *n* = 9, donor *n* = 12, recipient *n* = 12); (**C**) CD49a expression was studied and analyzed on the Trm cells (one-way ANOVA, * *p* < 0.05; Tukey’s multiple-comparison test * *p* < 0.05; CABG *n* = 11, donor *n* = 12, recipient *n* = 15). Representative flow cytometry images are shown.

**Figure 4 ijms-26-09852-f004:**
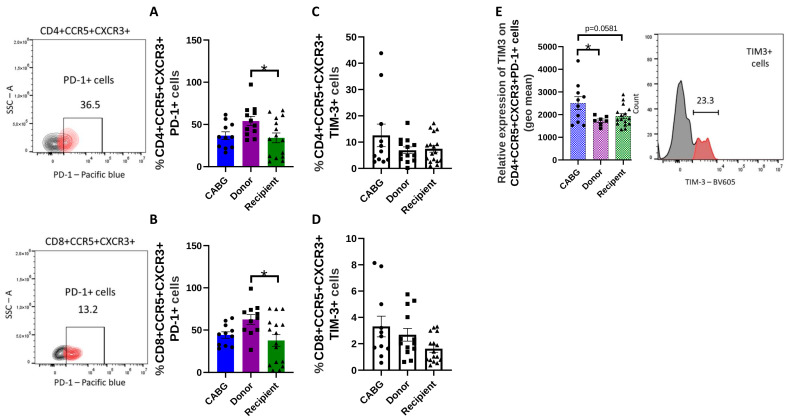
**Exhausted T cells in the pericardial fluid:** (**A**–**D**) PD-1 and TIM-3 expression was analyzed by flow cytometry in inflammatory heart-homing T cells, and the proportion of positive cells is presented (one-way ANOVA, * *p* < 0.05; Tukey’s multiple-comparison test * *p* < 0.05 A: CABG *n* = 11, donor *n* = 12, recipient *n* = 15; B: CABG *n* = 11, donor *n* = 11, recipient *n* = 16); (**E**) CD4^+^ CCR5^+^ CXCR3^+^ PD-1^+^ TIM3^+^ cells were gated in the flow cytometry plot, and the relative expression, determined by the geometric mean data, was calculated and visualized (one-way ANOVA, * *p* < 0.05; Tukey’s multiple-comparison test * *p* < 0.05; CABG *n* = 11, donor *n* = 7, recipient *n* = 16). Representative flow cytometry images are shown.

**Figure 5 ijms-26-09852-f005:**
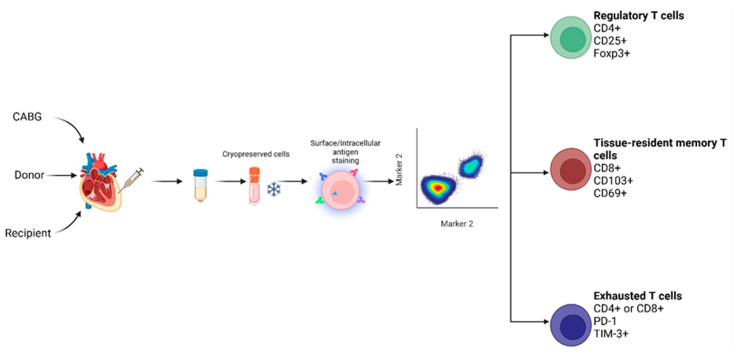
Workflow for sample collection and preparation.

**Figure 6 ijms-26-09852-f006:**
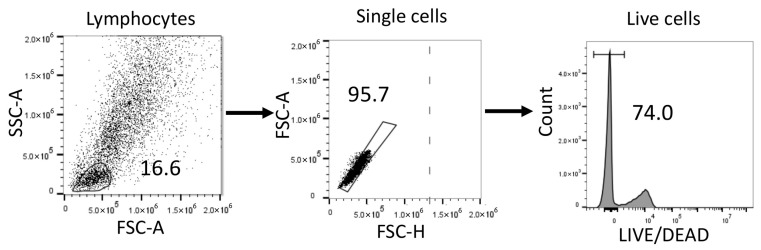
Flow cytometry gating strategy applied to all samples.

**Table 1 ijms-26-09852-t001:** Phenotype characteristics of CABG patients and recipients.

	Recipient Characteristics		CABG Patients Characteristics
**Age (year)**	51.00 (30.00–58.50)		69.00 (63.00–74.00)
**BMI (kg/m^2^)**	24.90 (21.00–25.79)		-
**Sex (Female/Male)**	5 (29.41%)/12 (70.59%)		3 (27.27%)/8 (72.73%)
**Comorbidities**	Hypertonia: 7 (42.18%) DM: 2 (11.76%) Hypothyreosis: 3 (17.65%)		Hyperlipidemia: 3 (27.27%) DM: 5 (45.45%) Hypothyreosis: 3 (27.27%)
**Etiology**	**DCM**	**Non-DCM**	**Ischemic Heart Disease**
	Familial: 3 (17.65%) Viral: 1 (5.88%) Ischemic: 2 (11.76%) Non-ischemic: 7 (42.18%)	Restrictive: 1 (5.88%) Ischemic: 1 (5.88%) HCM: 2 (11.76%)	11 (100%)
**Scores**	**UNOS recipient score**		**EuroSCORE**
	Very low risk: 2 (11.76%) Low risk: 8 (47.06%) Intermediate risk: 2 (11.76%) High risk: 2 (11.76%) Very high risk: 3 (17.65%)		2: 2 (18.18%) 4: 1 (9.09%) 5: 1 (9.09%) 6: 3 (27.27%)
	**IMPACT**		
	4.00 (1.25–5.00)		
**Classification**	**NYHA**		**CCS**
	II: 4 (23.53%) III: 3 (17.65%) IV: 10 (58.82%)		I: 1 (9.09%) II: 2 (18.18%) III: 2 (18.18%) IV: 2 (18.18%)
**Pre-op. laboratory values**			
CRP (mg/L) CK (IU/L) BUN (mmol/L) Creatinine (µmol/L) GFR (mL/min/1.73m^2^) AST (UI/L) ALT (UI/L) GGT (UI/L) ALP (UI/L) LDH (UI/L) Total Bilirubin (µmol/L) Total Protein (g/L) Albumin (g/L) Cholesterol (mmol/L) Triglycerides (mmol/L) HDL (mmol/L) LDL (mmol/L) PLT (G/L) Haemoglobin (g/L) WBC (G/L) Neutrophil % Neutrophil Count (10^9^/L) Lymphocyte % Lymphocyte Count (10^9^/L) Eosinophil % Eosinophil (10^9^/L) Monocyte % Monocyte Count (10^9^/L) Basophil % Basophil Count (10^9^/L)	2.35 (1.10–8.31) 115.50 (59.50–200.25) 9.90 (5.50–12.80) 114.00 (84.25–155.50) 79.89 (48.11–90.00) 23.00 (18.00–32.00) 29.00 (13.00–34.00) 81.00 (37.00–160.00) 77.00 (60.00–131.00) 417.50 (330.00–506.75) 11.00 (6.55–18.78) 71.50 (65.60–77.60) 46.50 (40.30–49.20) 3.80 (3.50–4.80) 1.18 (0.94–2.32) 1.36 (0.95–1.43) 2.92 (2.21–3.71) 206.00 (144.80–240.00) 130.50 (94.00–137.75) 8.74 (7.40–9.99) 73.00 (68.05–77.20) 5.76 (5.04–6.92) 16.10 (13.75–22.10) 1.25 (0.96–1.62) 1.20 (0.63–2.10) 0.11 (0.06–0.15) 7.85 (7.15–9.68) 0.72 (0.50–0.86) 0.35 (0.20–0.78) 0.03 (0.02–0.06)		2.31 (0.50–4.85) 94.50 (48.75–197.25) 6.50 (4.45–8.20) 70.50 (67.00–95.75) 88.60 (75.90–90.00) 21.00 (19.00–28.75) 31.00 (14.25–39.25) 27.50 (14.25–47.75) 81.00 (63.50–89.00) 316.00 (225.25–366.75) 9.20 (6.70–12.38) 69.60 (68.00–76.20) – 3.85 (3.08–4.83) 1.24 (0.93–1.69) 1.22 (0.95–1.44) 2.04 (1.57–3.16) 254.50 (176.75–279.75) 148.00 (136.00–158.00) 7.05 (6.24–7.56) 60.75 (56.75–68.10) 4.17 (3.94–4.62) 26.85 (21.68–30.98) 1.83 (1.55–2.16) 2.35 (1.85–4.83) 0.17 (0.11–0.34) 7.75 (6.80–8.35) 0.54 (0.41–0.60) 0.50 (0.50–0.68) 0.03 (0.03–0.05)
**Preoperative echocardiography** **parameters and hemodynamic** **measurement**	**Echocardiography parameters**		**Echocardiography parameters**
	EF (%): 22.50 (20.00–25.75)		EF (%): 56.5 (43.75–59.75) LVEDD (mm): 50.0 (42.0–50.0) LVESD (mm): 35.0 (28.0–38.0) LVPWd (mm): 10.0 (9.00–11.0) AscAoD (mm): 30.5 (29.0–32.0) IVSd (mm): 11.0 (9.0–14.0) Systolic AoD (mm): 21.0 (17.0–22.0) AoVmax (m/s): 1.30 (1.05–1.43) TAPSE (mm): 22.0 (20.0–24.0)
	**Hemodynamic measurement**		
	PVR (Wood Units): 2.40 (1.76–2.85) sPAP (mmHg): 45.00 (39.00–55.50) dPAP (mmHg): 22.00 (16.50–25.50) mPAP (mmHg): 33.00 (25.00–37.00) PAWP (mmHg): 22.00 (17.00–24.50)		
**Perioperative complications**	BTT: 3 (17.65%) Post-op MCS: 2 (11.8%) 1 year mortality: 0 (0%) 2 year mortality: 2 (11.8%) 5 year mortality: 2 (11.8%)	Vasoplegia: 4 (23.5%) PGD: 3 (17.6%) Rejection: 1 (5.9%) Reoperation: 1 (5.9%) Retransplant.: 0 (0%)	**-**
**Postoperative Inotropic Support**	Max-IS: 37.50 (16.10–66.38) Max-VIS: 41.00 (29.40–72.33) Norepinephrine equivalence: 0.31 (0.20–0.66)		-

Variables are presented as medians and interquartile (IQR25-75) ranges. Categorical data are presented as numbers and percentages. AoD: aortic root diameter, AscAoD: ascending aorta diameter, ALP: alkaline phosphatase, ALT: alanine transaminase, AST: aspartate aminotransferase, BMI: body mass index, BUN: blood urea nitrogen, BTT: bridge to transplant, CCS: Canadian Cardiovascular Society Classification, CK: creatine kinase, CRP: C-reactive protein, DCM: dilated cardiomyopathy, DM: diabetes mellitus, EuroSCORE: European System for Cardiac Operative Risk Evaluation, GFR: glomerular filtration rate, GGT: gamma-glutamyl transferase, HDL: high-density lipoprotein, IMPACT: Index for Mortality Prediction After Cardiac Transplantation, IVSd: interventricular septum thickness at end-diastole, LDH: lactate dehydrogenase, LVEDD: left ventricular end-diastolic diameter, LVESD: left ventricular end-systolic diameter, LVPWd: left ventricular wall end-diastole, Max-IS: maximum inotropic score, Max-VIS: maximum vasoactive–inotropic score, NEE: Norepinephrine equivalence, NYHA: New York Heart Association, PAWP: pulmonary artery wedge pressure, PGD: primary graft dysfunction, PLT: platelet count, PVR: pulmonary vascular resistance, sPAP: systolic pulmonary artery pressure, dPAP: diastolic pulmonary artery pressure, mPAP: mean pulmonary artery pressure, TAPSE: tricuspid annular plane systolic excursion, UNOS: United Network for Organ Sharing, WBCs: white blood cells.

**Table 2 ijms-26-09852-t002:** Phenotype characteristics of donors.

	Donor Characteristics
**Age (year)**	39.50 (31.50–47.50)
**BMI (** **kg/m^2^)**	26.40 (22.48–29.65)
**Sex (Female/Male)**	2 (14.29%)/12 (85.71%)
**Cause of death**	Intracerebral hemorrhage: 7 (50.00%) Traumatic subdural hemorrhage: 1 (7.14%) Traumatic subarachnoid hemorrhage: 1 (7.14%) Diffuse brain injury: 4 (28.57%) Anoxia: 1 (7.14%)
**UNOS donor risk score**	Low risk: 6 (42.86%) Intermediate risk: 8 (57.14%)

Variables are presented as medians and interquartile (IQR25–75) ranges. Categorical data are presented as numbers and percentages. BMI: body mass index, UNOS: United Network for Organ Sharing.

## Data Availability

The raw data supporting the conclusions of this article will be made available by the authors.

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
