# Peer review of "Pattern of Regulatory T Cells, Resident Memory T Cells, and Exhausted T Cells in Human Pericardial Fluid Samples of Cardiovascular Patients"

_ijms, 2025, doi:10.3390/ijms26209852_

Round 1
Reviewer 1 Report
Comments and Suggestions for Authors
- Unfortunately, the current Discussion section of the manuscript is overly concise, focusing mainly on a descriptive summary of the results. To enhance the academic depth and overall value of the paper, it is recommended to expand this section with more in-depth analysis, such as exploring underlying mechanisms, strengthening clinical relevance, and highlighting study limitations along with potential future directions.
- The references cited in the paper from the past 3-5 years are relatively few, which significantly undermines its frontier nature and innovativeness. It is recommended to systematically supplement the latest research in this field to strengthen the foundation of the argument.
- The description of Figures 2-5 in the Results section is rather brief and does not fully explain the biological significance of the findings shown. It is recommended to add one or two concluding statements to make it clearer and more straightforward.
- There are inconsistencies in the use of terminology in the manuscript, such as “Trm”and “TRM”. The abbreviations should be standardized to ensure consistent terminology.
Author Response
Reviewer 1:
Comment 1: “Unfortunately, the current Discussion section of the manuscript is overly concise, focusing mainly on a descriptive summary of the results. To enhance the academic depth and overall value of the paper, it is recommended to expand this section with more in-depth analysis, such as exploring underlying mechanisms, strengthening clinical relevance, and highlighting study limitations along with potential future directions.”
Response 1: We thank the reviewer for this valuable suggestion. In the revised version, the Discussion has been substantially expanded. We have integrated deeper mechanistic interpretations of our findings, strengthened the clinical relevance by referencing potential implications for heart failure and ischemic heart disease, and added new sections on study limitations and future directions. We believe these changes significantly improve the academic depth and translational value of the manuscript.
Comment 2: “The references cited in the paper from the past 3-5 years are relatively few, which significantly undermines its frontier nature and innovativeness. It is recommended to systematically supplement the latest research in this field to strengthen the foundation of the argument.
Response 2: We thank the review for this observation. The reference list has been systematically updated and expanded with recent publications from the last 3–5 years. These include novel insights into tissue-resident memory T cells, pericardial and epicardial immune niches, and inflammatory regulation in cardiac pathology. This ensures that the study is framed in the context of the most up-to-date research. Deniset et al. 2019 (Immunity), Huo et al. 2025 (Frontiers in Pharmacology), Wang et al. 2025 (Nature), Hassanabad et al. 2021 (JACC), Zhou et al. 2024 (Biomarker research).
Comment 3: “The description of Figures 2-5 in the Results section is rather brief and does not fully explain the biological significance of the findings shown. It is recommended to add one or two concluding statements to make it clearer and more straightforward.”
Response 3: We thank the reviewer for this helpful observation. We have revised the Results section for Figures 2–5 and added concluding statements to each subsection that highlight the biological significance of the findings. These additions provide clearer interpretations and improve the readability of the Results.
Comment 4: “There are inconsistencies in the use of terminology in the manuscript, such as “Trm”and “TRM”. The abbreviations should be standardized to ensure consistent terminology.”
Response 4: We appreciate the reviewer’s attention to consistency. All terminology and abbreviations have been carefully revised, and we now consistently use Trm throughout the manuscript.
Reviewer 2 Report
Comments and Suggestions for Authors
Introduction:
The section contains information about the relationship between the state of T cells and cardiovascular diseases. The authors describe the role of chemokine receptors in such a phenomenon as T-cell cardiotropism. The potential role of T cells in the pathophysiology of heart diseases is described.
It is necessary to decipher at the first use of Dilated Cardiomyopathy (DCM) - line 49
The role of T cells in DCM is described, as well as the features of T-reg distribution in heart diseases. The role of Tissue-resident memory T cells (Trms) r is described. In general, we see an adequate and logical introduction, where modern references are used.
However, at the end of the introduction section it is necessary to formulate the purpose of this study, both strategic and tactical, with the formulation of tasks.
Materials and methods.
2.1. Study Design, Setting and Participants
2.2. Variables and Definitions
2.3. Sample Collection and Preparation
The sections are well written, no comments or questions arose. Table 1 presents all the necessary information about the studied samples.
Ps the table has shifted - line 148
2.4. Flow Cytometry
The section is written adequately, the antibodies that were used in the work are indicated, as well as the key cell phenotypes:
Treg - CD4+CD25+Foxp3+
Trm - CD8+CD103+CD69+
Tex - CD4+ and CD8+ PD-1 and/or TIM-3+
It is worth noting that the supplement presents all the cell phenotypes that were analyzed.
2.5. Statistics – adequately described, all criteria and methods of analysis used are indicated.
In Figure 3, we see the design of the study, which increases the level of perception of the article.
- Results.
3.1. Enhanced T Cell Migration to the Heart in Transplantation Recipients
The data are presented in Figure 2 (A, B). The data are adequately presented, we can see individual values on the graph, as well as the gating tactics, the number of samples in groups and statistical criteria are indicated.
The text clearly describes the results.
It is nice to see such high-quality material presented, looking at which no additional questions arise).
3.2. Increased Infiltration, but Decreased Treg Function in Transplant Recipients
The data are presented in Figure 3. The data are presented adequately, we can see the individual values on the graph, as well as the gating tactics, the number of samples in the groups and the statistical criteria are indicated.
The text clearly describes the results.
3.3. Tissue Resident Memory T Cells (Trms) Are Enriched in Patients with DCM, but Their Homing and Retention May Be Impaired in the Pericardial Fluid
3.4. Reduced Number of Transiently Exhausted, but Increased Number of Terminally Exhausted T Cells in the Cardiac Patients
Similar to the previous points. The reviewer highly appreciates the way the data are presented and visualized.
The study used unique clinical material, so the article has a high level of interest to readers.
3.5. Clinical Variables
At this point, the authors refer to the sample because no significant associations were found between T-cell levels and complications, clinical assessments, echocardiographic parameters, hemodynamic measurements, or laboratory parameters among transplant recipients. No questions or comments were raised.
- Discussion
The authors analyze the data obtained in detail, explaining the results for each cell type. Given the fact that there are few studies on cardiotropism, the reviewer suggests supplementing the discussion with the following references, at the discretion of the authors:
Iskandar R, Liu S, Xiang F, Chen W, Li L, Qin W, Huang F, Chen X. Expression of pericardial fluid T-cells and related inflammatory cytokines in patients with chronic heart failure. Exp Ther Med. 2017 May;13(5):1850-1858. doi: 10.3892/etm.2017.4202.
The article showed inflammatory cell infiltration and increased expression of inflammatory cytokines in the pericardial fluid of patients with CHF, which may reflect T cell activation, suggesting that systemic inflammation plays an important role in the progression of CHF.
Alexander MR, Dale BL, Smart CD, Elijovich F, Wogsland CE, Lima SM, Irish JM, Madhur MS. Immune Profiling Reveals Decreases in Circulating Regulatory and Exhausted T Cells in Human Hypertension. JACC Basic Transl Sci. 2023 Jan 4;8(3):319-336. doi: 10.1016/j.jacbts.2022.09.007.
The study results provide new evidence for a reduction in anti-inflammatory and/or hypofunctional T cell populations, which may contribute to the increased inflammation in hypertension in humans.
The reviewer has no personal benefit from these recommendations.
Thus, the main comment concerns the need to formulate the purpose of the study at the end of the introduction section, decipher the DCM, and supplement the discussion with recommended references (at the discretion of the authors).

Author Response
Reviewer 2
Comment 1: “It is necessary to decipher at the first use of Dilated Cardiomyopathy (DCM) – line 49.”
Response 1: We thank the reviewer for this observation. The abbreviation has now been deciphered at its first mention in the Introduction.
Comment 2: “However, at the end of the introduction section it is necessary to formulate the purpose of this study, both strategic and tactical, with the formulation of tasks.”
Response 2: We fully agree. The Introduction has been revised to include a clear statement of the study’s purpose and objectives at the end of the section.
Comment 3: “The reviewer suggests supplementing the discussion with the following references, at the discretion of the authors: Iskandar et al., 2017; Alexander et al., 2023.”
Response 3: We thank the reviewer for these valuable recommendations. Both references have been incorporated into the paper. The study by Iskandar et al. (2017) is cited in relation to the pro-inflammatory profile of pericardial fluid in chronic heart failure, while the study by Alexander et al. (2023) is cited when discussing the distribution of regulatory and exhausted T-cell subsets in cardiovascular disease. We believe their inclusion strengthens the clinical context and supports our interpretations.
Reviewer 3 Report
Comments and Suggestions for Authors
The results of this study are of interest due to the fact that it was conducted on unique material.
The following minor and major comments should be addressed:
Major
- There is a missing conclusion at the end of the introductory section.
- Figures 2,3,4,5 - The data is given in percentages, but it is not clear what percentage of cells is being referred to - lymphocytes, CD4+ cells, etc.
- The absolute numbers of cells are not presented.
- The method for isolating live and dead cells should be described in the methods section.
- Pericardial fluid is the result of plasma ultrafiltration from the epicardial capillary bed, pericardial mesothelial cells and myocardial interstitial space Pericardial fluid contains a heterogeneous population of cells, including epithelial cells, lymphocytes, neutrophils, macrophages, as well as eosinophils and basophils. The methods section does not provide information about the amount of pericardial fluid collected, the cell composition of pericardial fluid, or the average number of cells obtained. Therefore, these details should be included in the limitations section of the paper. Further discussion on this topic is also needed. Pericardium is a double-walled sac. Why do the authors suggest that lymphocytes found in the pericardial fluid migrate to the myocardium, or are they coming from the myocardium itself?
- Authors write “Directed heart homing was considered when cells were double positive for CCR5 and CXCR3”, “In the context of T cell cardiotropism, where CCR5 and CXCR3 (4, 6, 7) are the primary migratory receptors….” The expression of these receptors does not indicate cardiotropism. Instead, it means that the cells will move to the site of inflammation, where there are specific chemokines present. Th1 cells may have such a phenotype and can be identified in atherosclerotic plaques. The expression of CCR4 on Treg cells does not indicate cardiac specificity; Th2 cells express the CCR4 too. The selected panel for immunophenotyping needs justification. These points should be added to the limitations section. Additionally, the discussion on this topic should be expanded. Similarly, based on the selected antibody panel the statements ‘CCR4+ specifically heart homing CD4+CD25+Foxp3+ Treg cell’, ‘homing T cells’ are not entirely accurate.
- The authors report “A strong negative correlation was found between TAPSE and Treg CD4+FOXP3+CCR5+Q4:CCR4-, CXCR3- levels (r = –0.991; p = <0.001)” (line 310). However, they do not provide any detailed information about these cells or discuss the correlation in depth.
- Figure 5. The authors should carefully check the captions for the figure (A,B,C,D in the title, but A,B,C,D,E in the figure; the letters must be in the correct position in relation to the drawing). 5E(?) Y axis – it is not clear the relative expression of what is presented (TIM3?).
Minor
1) The abbreviation 'HTx' is used only once, so it should be defined.
2) It is recommended to include a brief description of the donor's gender, age and cause of death.
3) The manufacturer of the antibodies is Sony Biotechnology Inc., and not Sony itself. Listing the manufacturer after each antibody is not necessary.
4) Figure 1 is not very informative. Everything is clear from the text.
5) The methodology lists antibodies for CD45, 49a, and 127. However, the results are not presented.
6) The example of separating the lymphocyte window and gating living cells should be transferred the methods section.
7) The authors report that they included 11 donors, 16 recipients and 9 CABG patients (methods section).
According to Table 1, there were 9 CABG patients and 17 total recipients.
According to Figure 2: CABG n=10, Donor n=13, Recipient n= 15.
According to Figure 3: CABG n=11, Donor n=13, Recipient n=15.
According to Figure 4: CABG n=11, Donor n=10, Recipient n=16.
The authors should carefully check the total number of patients in each analyzed group.
8) The two tables on CABG patients and recipients in Table 1 should be merged into one. The table needs to be formatted in a way that makes it more compact. For recipients, lymphocyte and neutrophil data are provided. For patients with CABG more detailed laboratory data is provided, including monocytes, basophils, eosinophils.
9) It is not clear what does "from 4 independent experiments" mean in a cross-sectional study design? The same cells were stained 4 times?
English style and grammar corrections are needed.
Author Response
Reviewer 3:
Comment 1: “There is a missing conclusion at the end of the introductory section.”
Response 1: We thank the reviewer for this helpful suggestion. In the revised manuscript, we have added a clear concluding paragraph at the end of the Introduction that outlines the purpose of the study and the specific objectives.
Comment 2: “Figures 2,3,4,5 - The data is given in percentages, but it is not clear what percentage of cells is being referred to - lymphocytes, CD4+ cells, etc.”
Response 2: We thank the reviewer for this helpful observation. The y-axis labels of Figures 2–5 have been revised to clearly indicate the reference population. We hope that this clarification improves the readability and interpretation of the figures.
Comment 3: “The absolute numbers of cells are not presented.”
Response 3: We thank the reviewer for this important point. In our study, we focused specifically on immunophenotyping of defined T-cell populations rather than on total cellularity of the pericardial fluid. For this reason, we did not determine or report absolute numbers of all nucleated cells, as our gating strategy was designed to resolve relative distributions within the lymphocyte compartment.
We acknowledge, however, that pericardial fluid cell counts can vary substantially between individuals. Previous studies have reported wide ranges of nucleated cell concentrations in PF: Fender et al. established reference intervals of 278–5,608 ×10⁶ cells/L in normal human samples (Heart. 2021 Jul 7;107(19):1528–1529.) while Buoro et al. described leukocyte concentrations between 3.8–6.0 ×10⁶ cells/L in clinical PF samples (Heart. 2021 Oct;107(19):1584-1590.). These values demonstrate the heterogeneity of pericardial fluid cellularity.
To minimize this source of variation, we did not include samples with very low cell counts in our analysis. Our results therefore represent relative changes in defined T-cell subsets under comparable cellular conditions. However, we agree that integration of absolute counts with immunophenotyping would provide additional information in certain contexts, and in our future analyses we will remember this point.
Comment 4: “The method for isolating live and dead cells should be described in the methods section.”
Response 4: We thank the reviewer for this comment. The isolation of live and dead cells was not performed through mechanical separation but rather by applying a fixable Live/Dead viability dye during flow cytometry analysis. Specifically, we used Fixable Live/Dead Scarlet (Thermo Fisher) to discriminate viable from non-viable cells, which is also illustrated in the gating strategy now shown in Figure 6. We have now supplemented the Methods section with a more explicit description of this step.
Comment 5: “The methods section does not provide information about the amount of pericardial fluid collected, the cell composition of pericardial fluid, or the average number of cells obtained. Therefore, these details should be included in the limitations section of the paper. Further discussion on this topic is also needed.”
Response 5: We thank the reviewer for this valuable suggestion. The Methods section has been updated to include information on the volume of pericardial fluid collected. In addition, the end of the Discussion section we included the limitations of our study, such as the lack of systematic data on the precise cellular composition and average cell yield from pericardial fluid samples.
Comment 6: “Why do the authors suggest that lymphocytes found in the pericardial fluid migrate to the myocardium, or are they coming from the myocardium itself?”
Response 6: We thank the reviewer for this insightful question. The presence of lymphocytes in pericardial fluid likely reflects a bidirectional process: cells may drain into the fluid from inflamed myocardium, while others may also migrate inward from the pericardial compartment toward the heart in response to chemokine gradients. Experimental evidence supports such dynamics: for example, Gata6⁺ macrophages residing in the pericardial cavity have been shown to relocate into the injured myocardium and limit fibrosis (Deniset et al. Immunity. 2019 Jul 16;51(1):131-140.e5). Although our data cannot resolve the directionality, the observed enrichment of inflammatory CCR5⁺CXCR3⁺ T cells in failing hearts is compatible with active communication between pericardial space and myocardium. The Discussion has been revised to include this point.
Comment 7: “Authors write “Directed heart homing was considered when cells were double positive for CCR5 and CXCR3”, “In the context of T cell cardiotropism, where CCR5 and CXCR3 (4, 6, 7) are the primary migratory receptors….” The expression of these receptors does not indicate cardiotropism. Instead, it means that the cells will move to the site of inflammation, where there are specific chemokines present. Th1 cells may have such a phenotype and can be identified in atherosclerotic plaques. The expression of CCR4 on Treg cells does not indicate cardiac specificity; Th2 cells express the CCR4 too. The selected panel for immunophenotyping needs justification. These points should be added to the limitations section. Additionally, the discussion on this topic should be expanded. Similarly, based on the selected antibody panel the statements ‘CCR4+ specifically heart homing CD4+CD25+Foxp3+ Treg cell’, ‘homing T cells’ are not entirely accurate.”
Response 7: We thank the reviewer for this constructive comment and agree that CCR5, CXCR3, and CCR4 are not cardiac-exclusive receptors. We selected these markers because several studies demonstrate that they form part of a broader “heart-homing program” under inflammatory conditions. In particular, hepatocyte growth factor (HGF) produced by the heart can instruct T cells during priming via the c-Met receptor, creating a cardiotropic signature that includes c-Met together with CCR4 and CXCR3 (Komarowska et al. Immunity. 2015 Jun 16;42(6):1087-99.). CXCR3 ligands such as CXCL10 are strongly upregulated in failing myocardium and promote T-cell infiltration (Melter et al. Circulation. 2001 Nov 20;104(21):2558-64.), while CCR4 ligands CCL17/22 can regulate Treg trafficking to the injured heart (Feng et al. Circulation. 2022 Mar 8;145(10):765-782.). Furthermore, CCR5 and CXCR3 are critical for CD4⁺ T-cell infiltration and remodeling in cardiac stress models (Ngwenyama et al. JCI Insight. 2019 Apr 4;4(7):e125527.). Taken together, these findings may justify our choice of CCR5, CXCR3, and CCR4 as valid and biologically relevant markers of T-cell migration in the cardiac inflammatory context. We appreciate this thoughtful suggestion and have refined our terminology throughout the manuscript.
Comment 8: “The authors report “A strong negative correlation was found between TAPSE and Treg CD4+FOXP3+CCR5+Q4:CCR4-, CXCR3- levels (r = –0.991; p = <0.001)” (line 310). However, they do not provide any detailed information about these cells or discuss the correlation in depth.”
Response 8: We thank the reviewer for this comment. We have expanded the relevant section and we discuss the possible biological implications of the latter, while also noting that all CABG patients had TAPSE values within the normal range. The limitations of this finding are emphasized, and we state that larger cohorts will be required to clarify this relationship.
Comment 9: “Figure 5. The authors should carefully check the captions for the figure (A,B,C,D in the title, but A,B,C,D,E in the figure; the letters must be in the correct position in relation to the drawing). 5E(?) Y axis – it is not clear the relative expression of what is presented (TIM3?).”
Response 9: We thank the reviewer for pointing this out. The figure panels and captions for Figure 5 have been carefully revised to ensure consistent lettering, and the y-axis labeling in panel E has been corrected for clarity.
Minor
Comment 10: “1) The abbreviation 'HTx' is used only once, so it should be defined.”
Response 10: The abbreviation “HTx” has been defined at first use in the manuscript.
Comment 11: “2) It is recommended to include a brief description of the donor's gender, age and cause of death.”
Response 11: The phenotype characteristics of donors gender, age and cause of death has been added to the manuscript in Table 2.
Comment 12: “3) The manufacturer of the antibodies is Sony Biotechnology Inc., and not Sony itself. Listing the manufacturer after each antibody is not necessary.”
Response 12: We corrected the manufacturer’s name.
Comment 13: “4) Figure 1 is not very informative. Everything is clear from the text.”
Response 13: With all due respect, we would like to respectfully disagree with the reviewer's comment. We believe that this figure is valuable, as it helps readers quickly understand the experimental strategy used to identify T cell subsets.
Comment 14: “5) The methodology lists antibodies for CD45, 49a, and 127. However, the results are not presented.”
Response 14: We thank the reviewer for noting this point. Indeed, we used CD45RA in the analysis of Treg subsets and CD49a in defining Trm cells, and these results are presented in the manuscript. CD127, however, was indeed not included in the analyses, and we have now removed it from the antibody list in the Methods.
Comment 15: “6) The example of separating the lymphocyte window and gating living cells should be transferred the methods section.”
Response 15: As suggested, we carried out this modification.
Comment 16: “7) The authors report that they included 11 donors, 16 recipients and 9 CABG patients (methods section).
According to Table 1, there were 9 CABG patients and 17 total recipients.
According to Figure 2: CABG n=10, Donor n=13, Recipient n= 15.
According to Figure 3: CABG n=11, Donor n=13, Recipient n=15.
According to Figure 4: CABG n=11, Donor n=10, Recipient n=16.
The authors should carefully check the total number of patients in each analyzed group.”
Response 16: We thank the reviewer for this careful observation. We have reviewed and corrected the numbers across the manuscript. Unfortunately, in a few instances, the number of samples is lower than the total cohort due to technical issues that prevented all specimens from being analyzed with every staining panel.
Comment 17: “8) The two tables on CABG patients and recipients in Table 1 should be merged into one. The table needs to be formatted in a way that makes it more compact. For recipients, lymphocyte and neutrophil data are provided. For patients with CABG more detailed laboratory data is provided, including monocytes, basophils, eosinophils.”
Response 17: As suggested, we carried out this modification.
Comment 18: “9) It is not clear what does "from 4 independent experiments" mean in a cross-sectional study design? The same cells were stained 4 times?”
Response 18: We thank the reviewer for raising this point. We understand that the term “4 independent experiments” is confusing here, and therefore, we removed this term from the text. We originally referred to the fact that we measured the human samples on four different days in four batches.
Comment 19: “English style and grammar corrections are needed.”
Response 19: We thank the reviewer for this comment. The entire manuscript has been carefully reviewed for English language and grammar, and we have revised the text accordingly. We hope that the readability and style are now improved.
Round 2
Reviewer 1 Report
Comments and Suggestions for Authors
The author has completed a second round of revisions to the manuscript, addressing previous comments and incorporating recent references to strengthen the scholarly foundation. Significant improvements include a more critical Discussion section, with explicit acknowledgment of limitations and future research directions, as well as a clearer and more concise Results section. However, some issues in spelling, data interpretation, and structural clarity still require further refinement.
1. In the statistical chart in Figure 1B, the group name “Rrecipient” is misspelled. Please check carefully and correct it.
2. The sample numbers in the statistical charts are inconsistent with the description in the legend. Please check the original data and figures to ensure that the sample numbers are correct. For example, in Figure 1A, the Donor group is labeled as n=12, but there are actually 14 points in the chart; in Figure 1B, the Recipient group is labeled as n=15, but only 13 points are shown in the chart.
Author Response
Comment 1:
In the statistical chart in Figure 1B, the group name “Rrecipient” is misspelled. Please check carefully and correct it.
Response 1:
We thank you for pointing this error. The label has been corrected to “Recipient” in Figure 1B.
Comment 2:
The sample numbers in the statistical charts are inconsistent with the description in the legend. Please check the original data and figures to ensure that the sample numbers are correct. For example, in Figure 1A, the Donor group is labeled as n=12, but there are actually 14 points in the chart; in Figure 1B, the Recipient group is labeled as n=15, but only 13 points are shown in the chart.
Response2:
We thank the reviewer for carefully noting this point. We have re-checked the original datasets and figures and clarified the discrepancies. In some instances, the inconsistency was due to a labeling error in the figure legend, where more samples were included in the analysis than originally stated. In other cases, a small number of samples could not be analyzed in specific panels due to technical issues (e.g., failed staining or insufficient signal). Two donor samples were used only for preliminary setup and gating strategy optimization and were therefore not included across all panels. All sample numbers have now been corrected in the figure legends to accurately reflect the data shown.
Reviewer 3 Report
Comments and Suggestions for Authors
Line 164 – the preposition "an" is not necessary.
Line 183 – ‘The overall number of Trm cells…’ – The term "overall number" is primarily relevant to the absolute counts of cells. However, as the authors have calculated the relative content of cells, it would be more appropriate to use % or proportions when describing these values.
Lines 287-290 – ‘When considered together, our pericardial enrichment and the hypertension-associated circulating deficit suggest a model in which cardiovascular inflammation differentially partition T-cell subsets between blood and cardiac tissues depending on the context of the disease and the chemokine environment.’ – in this sentence the English grammar should be carefully checked.
As T-cell populations were analyzed within lymphocyte population and presented in relative content (% lymphocytes), please add this information to the manuscript. For example, it could be added to the ‘Flow cytometry’ part in the Methods section. It is important to note that the data may be calculated as a percentage of lymphocytes, % of CD4+ T-cells (or other major cell populations) or in absolute numbers (considering the total number of leukocytes and lymphocytes).
Author Response
Comment 1 (Line 164): “The preposition an is not necessary.”
Response 1: We thank the reviewer for pointing this out. The redundant preposition has been removed in the revised manuscript.
Comment 2 (Line 183): “‘The overall number of Trm cells…’ – The term overall number is primarily relevant to the absolute counts of cells. However, as the authors have calculated the relative content of cells, it would be more appropriate to use % or proportions when describing these values.”
Response 2: We appreciate this important clarification. The wording has been revised accordingly, and we now describe these values in proportions.
Comment 3(Lines 287–290): “‘When considered together, our pericardial enrichment and the hypertension-associated circulating deficit suggest a model in which cardiovascular inflammation differentially partition T-cell subsets between blood and cardiac tissues depending on the context of the disease and the chemokine environment.’ – in this sentence the English grammar should be carefully checked.”
Response 3: We thank the reviewer for noting this. The sentence has been carefully revised for clarity and correctness in the updated manuscript.
Comment 4: “As T-cell populations were analyzed within lymphocyte population and presented in relative content (% lymphocytes), please add this information to the manuscript. For example, it could be added to the ‘Flow cytometry’ part in the Methods section. It is important to note that the data may be calculated as a percentage of lymphocytes, % of CD4+ T-cells (or other major cell populations) or in absolute numbers (considering the total number of leukocytes and lymphocytes).”
Response 4: We agree with the reviewer’s suggestion. This information has now been added to the Flow cytometry section in the Methods, where we specify that T-cell populations were analyzed within the lymphocyte.